# Magnetic Properties of NiZn Ferrite Nanofibers Prepared by Electrospinning

**Kyeong-Han Na [1], Wan-Tae Kim [1], Tae-Hyeob Song [2] and Won-Youl Choi [1,3,*]**

[1] Department of Advanced Materials Engineering, Gangneung-Wonju National University, Gangneung 25457, Korea; nag0717@naver.com (K.-H.N.); dktkzz1@naver.com (W.-T.K.)
[2] Korea Institute of Civil Engineering and Building Technology, Goyang 10223, Korea; thsong@kict.re.kr
[3] Research Institute for Dental Engineering, Gangneung-Wonju National University, Gangneung 25457, Korea
[*] Correspondence: cwy@gwnu.ac.kr



**Featured Application: Filler for the electromagnetic-interference (EMI) shielding film.**

**Abstract:** When the size of a material is decreased to the nanoscale, the effects of forces that are not influential on a macroscopic scale become increasingly important and the electronic structure is improved. The material then exhibits significantly different physical and chemical properties than in the bulk state. The smaller the size of the material, the more exposure it receives to the nano effects, and the physical properties can be changed via size control. In this study, $Ni_{0.5}Zn_{0.5}Fe_2O_4$ ferrite nanofibers were prepared by electrospinning, and the sizes of the prepared samples were controlled to ensure different average diameters by controlling the polymer concentration of the precursor solution. Field emission scanning electron microscope images showed that the samples had average diameters of 224 to 265 nm. The single crystal phase of $Ni_{0.5}Zn_{0.5}Fe_2O_4$ and the different crystallite sizes of 13 to 20 nm were confirmed by X-ray diffraction analysis. The magnetization behavior of the samples was measured using a vibrating sample magnetometer and the result confirmed that the samples had different magnetic properties, according to the diameter and crystallite size of the nanofibers. This study suggests that control of magnetic properties and excellent electrical conductivity in a one-dimensional nanostructure can be positively applied to improve the performance of a filler for the electromagnetic-interference shielding film.

**Keywords:** electrospinning; NiZn ferrite; nanofibers; microstructure; magnetization behavior

## 1. Introduction

Electronic devices occupy a very important place in modern daily life and the industry. The circuits for these devices have become more portable and functional through increasingly higher levels of integration and complicated structure. Since this trend inevitably makes devices vulnerable to electromagnetic interference, there is a growing demand for simple and efficient electromagnetic wave shielding materials to prevent unnecessary wireless noise. Various studies have reported that, to achieve a thin and light shielding film, a magnetic ceramic filler is more potent than a heavy metal or pyramid-shaped shielding material. Transition metal oxides containing iron have unique electromagnetic, thermal, and mechanical properties, and various studies are underway in areas such as battery anode materials [1–3], catalyst materials [4,5], and sensors [6,7]. In particular, soft magnetic spinel ferrite has suitable magnetic properties for a ceramic paint filler to cope with high-frequency environments.

In recent years, there have been many attempts to improve the efficiency of shielding films and some of the methods reported were as follows. Substitute various elements for the bivalent Fe sites in



the $Fe_3O_4$ spinel ferrite structure [8–11], control the content of substituted or doped elements [12,13], and optimize the thickness of the shielding film [14,15]. On the other hand, excellent characteristics of a one-dimensional structure like porosity, a large specific area, and improved charge mobility have been attracting more attention because of their potential value in various applications, such as solar cells [16,17], photocatalysts [18–20], biosensors, and medical tissues [21–23]. Since the improved carrier mobility is considered to have an application value in the field of electromagnetic wave shielding film, several fillers with a one-dimensional nanostructure have been proposed [24–26]. To produce a one-dimensional nanostructure, methods such as hydrothermal [27,28], template [29], and anodic oxidation [30,31] are mainly used, and electrospinning have an advantage in that they can fabricate a uniform nanofiber rapidly even with a simple apparatus configuration [32]. At the same time, since the magnetization behavior of the material is closely related to the shielding efficiency and the use of the frequency band, many attempts have been made to control it. Research has focused not only on applying the one-dimensional nanostructure for a geometric advantage but also on optimizing the magnetic behavior by controlling the crystallite size. Some conventional studies have attempted to achieve this goal by increasing or decreasing the crystallite size or nanofiber diameter by controlling the heat treatment temperature [33–35]. NiZn ferrites are known to have high saturation magnetization and chemical stability, and attempts have been made to produce NiZn ferrite nanofibers and apply them as microwave absorbers [8,36]. However, there are no reports on how the magnetization behavior in NiZn ferrite nanofibers can be changed by complex microstructural changes, such as diameter and crystallite size resulting from the polymer content in the electrospinning precursor solution.

In this study, we fabricated NiZn ferrite nanofibers by the electrospinning method and controlled the microstructures, the crystallite size, and the nanofiber diameter, by polymer content in the precursor solution. As-spun nanofibers with a mixed composition of transition metal and polymer were calcined at suitable conditions and the calcined oxide nanofibers were investigated to explain the magnetization behavior of the nanofibers with various microstructures. The oxide nanofibers created different average diameters by controlling the polymer content, and the average diameter difference was measured by field emission scanning electron microscope (FE-SEM) image analysis. The presence of the same spinel crystal phase in each sample was confirmed by X-ray diffraction analysis (XRD) and a vibrating sample magnetometer (VSM) measured the magnetization behavior, according to the average diameter change of the nanofibers. Along with the doped and substituted elements, the particle and crystallite size of the structures were the major factors that governed the magnetic properties of the material. Utilizing this point, we found that the frequency band and efficiency of the shielding material could be finely controlled.

## 2. Materials and Methods

### 2.1. Fabrication of NiZn Ferrite Nanofibers via Electrospinning

The nanofibers were prepared by a typical electrospinning process as follows. $Ni(NO_3)_2 \cdot 6H_2O$ (13478-00-7, 98%, EP, Samchun Chemicals, Seoul, Korea), $Zn(NO_3)_2 \cdot 6H_2O$ (10196-18-6, 98%, EP, Daejung Chemicals, Siheung, Korea), and $Fe(NO_3)_2 \cdot 9H_2O$ (7782-61-8, 99%, GR, Kanto Chemical, Tokyo, Japan) in a molar ratio of 1:1:4 were mixed in N,N-dimethylmethanamide (DMF, 68-12-2, 99.5%, EP, Daejung chemicals, Siheung, Korea) as a solvent at a 150 mmol/L of molar concentration. The nitrate crystals were mixed using a magnetic stirrer until they were completely dissolved in DMF solvent. Polyvinyl pyrrolidone (PVP, 9003-39-8, 98.3%, M.W. 1,300,000, Alfa Aesar Co., Tewsbury, MA, US) was added to the completely dissolved solution at a weight ratio of 18 wt%, 20 wt%, and 22 wt% and stirred for 48 h to prepare the electrospinning precursor solution. The precursor solutions were labeled A, B, and C, and the compositions of the precursors are shown in Table 1. The prepared precursor was placed in a polypropylene syringe, connected with polyethylene tubing, and mounted on a syringe pump. A stainless-steel nozzle adapter and a 27-gauge plastic nozzle were attached to the end of the tubing, and pressure was applied through the pump to infuse at a constant flow rate of 0.5 mL/h. The nozzle

adapter was connected to the power supply, and the aluminum foil collector was grounded. A voltage of 15 kV was applied at the electrode while maintaining a 15-cm electrode-to-electrode distance. The relative humidity was 40% for room temperature. The collected composite nanofiber sheets were dried in a dry oven at 80 °C for 2 h. The dried composite nanofiber sheet was heated to a temperature of 750 °C at a rate of 5 °C/min atmospheric environment, fixed for 4 h, and then cooled to obtain a ceramic nanofiber with a $Ni_{0.5}Zn_{0.5}Fe_2O_4$ ferrite single phase. The morphologies and diameters of the nanofiber samples were confirmed by field emission-scanning electron microscopes (FE-SEM) image analysis. The heat treatment temperature of the composite fiber was determined by confirming the pyrolysis behavior using thermogravimetric analysis. The crystal phase was determined by comparing the XRD pattern with the Crystallography Open Database. VSM measured the hysteresis loop of each average diameter sample.

**Table 1.** The composition and content of the precursor solution. PVP: Polyvinyl pyrrolidone; DMF: N,N-dimethylmethanamide.

| Precursors | $Ni(NO_3)_2 \cdot 6H_2O$ (g) | $Zn(NO_3)_2 \cdot 6H_2O$ (g) | $Fe(NO_3)_2 \cdot 9H_2O$ (g) | PVP (g) | DMF (g) |
|---|---|---|---|---|---|
| Precursor A | | | | 18 | |
| Precursor B | 1.45 | 1.48 | 4.04 | 20 | 100 |
| Precursor C | | | | 22 | |

## 2.2. Characterization

Graphs of pyrolysis behavior of PVP/metal nitrate composite nanofibers were obtained by thermo-gravimetric analysis (TGA, Q500, TA Instruments Co., New Castle, DE, US). The weight change was measured while heating from 20 °C to 750 °C in air at a heating rate of 5 °C/min.

The crystal structure of the NiZn ferrite nanofibers after calcination was analyzed with an XRD diffraction pattern. XRD analysis was carried out by a diffractometer (AXS-D8, Bruker Co., US) and measured at 40 kV – 40 mA current conditions with a Cu K$\alpha$ radiation source. The measurements were performed at an analysis rate of 0.02°/s from 2θ = 20° to 80°.

The surface morphology and diameter of the calcined NiZn ferrite nanofiber samples were observed by FE-SEM (SU-70, Hitachi Co., Niigata, Japan). Images were collected at an applied voltage of 30 kV and a working distance of 10 mm. In addition, energy dispersive X-ray spectroscopy (EDX, Octane Elite EDS system, EDAX Inc., Mahwah, NJ, USA) and elemental mapping were carried out to confirm the distribution of the elements in the NiZn ferrite nanofibers.

The magnetization behavior of each sample was analyzed at room temperature through VSM. The applied magnetic field was measured from −15 kOe to 15 kOe, the data point was measured once per 500 Oe, the time constant was 0.1, and the analysis time was 1 s/pt.

## 3. Results

For the heat treatment process, Figure 1 shows the pyrolysis behavior graphs of the PVP/metal nitrate composite nanofiber sheet that were obtained by TGA. In the curve representing the weight to temperature, the weight of about 15 wt% sharply decreased until the temperature of the sample reached about 73 °C. This was due to the loss of moisture absorbed during the process. From this fact, the required temperature of the drying process was calculated at 80 °C immediately after the electrospinning process. The gradual loss curve from 150 °C to 420 °C indicates a range corresponding to the decomposition of PVP and the volatilization of the residual solvent. The mass loss range of pure PVP is 300–600 °C, but the decomposition temperature range of the metal salt and hybrid electro-spun nanofiber sheets shifted to a slightly lower temperature [37]. It is generally considered that the high thermal conductivity of the metal ions between the polymer chains promotes thermal decomposition. At 420 ° C, the mass reduction decreases and then begins to decrease again at 570 °C, which is the point where the NiZn ferrite fiber begins to crystallize into the spinel structure [38]. The reaction ends

at 685 °C with a final mass loss of 88 wt%. Regarding these experimental results, the heat treatment temperature of the nanofibers used in the analysis was determined to be 750 °C.

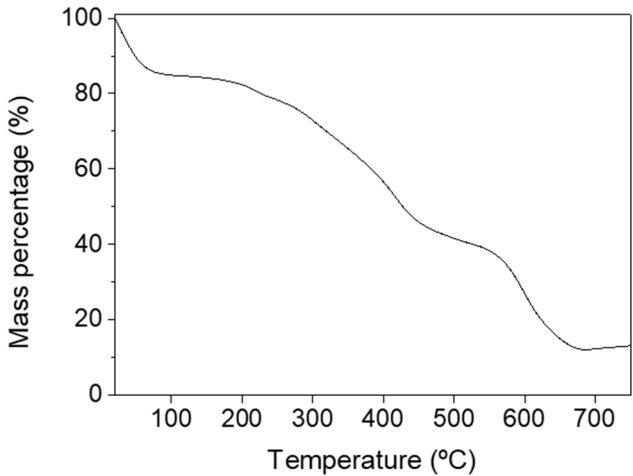

**Figure 1.** TGA curve of PVP/metal nitrate composite nanofibers.

The electro-spun samples from precursors A, B, and C were labeled and analyzed as samples A, B, and C. XRD analysis was used to identify the crystalized structure of the final samples. Figure 2 shows the diffraction patterns of samples A, B, and C calcined at 750 °C for 4 h. Despite the differences in polymer content, the diffraction pattern for the crystalized structure was the same for all three samples. The significant diffraction peaks indexed as (202), (131), (040), (242), (151), (404), and (444), which confirmed that the spinel structure was properly formed. The diffraction pattern was indexed by comparing with the 900-9920 entry of the Crystallography Open Database.

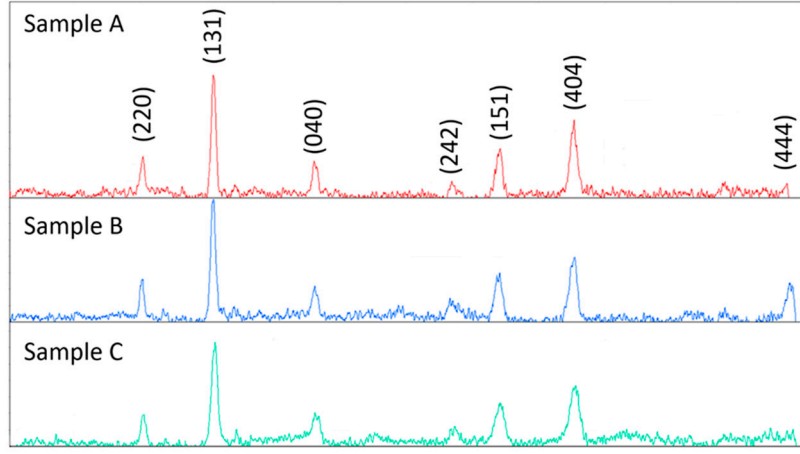

**Figure 2.** X-ray diffraction patterns of the PVP/metal nitrate composite nanofibers calcined at 750 °C.

Table 2 shows the results obtained by calculating the XRD diffraction pattern using the Scherrer formula (d = 0.94λ / B cos θ). In this formula, *d* represents the crystallite size, B is the full width at half maximum (FWHM) of each XRD peak, and $\lambda$ is the Cu K$_\alpha$ wavelength (0.15478 nm). The table shows the average crystallite size obtained using the peak intensities at (131), (151), and (404) diffraction planes per sample. The crystallite size calculated by the Scherer formula tends to decrease with an increasing PVP content of the precursor solution. According to Xiaolei Song et al., nanofibers with high polymer content in the calcination process increase the size and number of pores and cracks. These defects prevent the growth of a nuclear crystal and make the diffusion in the boundary of primary crystallites difficult [39]. The difference in crystallinity can also be confirmed by the width of the FWHM.

**Table 2.** Average crystallite size and XRD peak broadening data of each sample. FWHM: full width at half maximum.

| Samples | Plane (*hkl*) | 2θ (°) | FWHM (°) | Average Crystallite Size (nm) |
|---|---|---|---|---|
| Sample A | (131) | 35.25 | 0.33 | 20 ± 6 |
| | (151) | 56.72 | 0.58 | |
| | (404) | 62.47 | 0.56 | |
| Sample B | (131) | 35.60 | 0.44 | 16 ± 3 |
| | (151) | 56.76 | 0.69 | |
| | (404) | 62.82 | 0.68 | |
| Sample C | (131) | 35.63 | 0.58 | 13 ± 2 |
| | (151) | 56.78 | 0.87 | |
| | (404) | 62.81 | 0.82 | |

Once all the components except for the metallic element are volatilized during calcination, the fibrous structure may not be maintained even if a suitable crystalized structure is formed. Therefore, it is important to check the morphology through a microscopic observation. Figure 3a, Figure 3b, and Figure 3c are FE-SEM images of the as-spun nanofibers of samples A, B, and C, respectively, and Figure 3d, Figure 3e, and Figure 3f show the calcined $Ni_{0.5}Zn_{0.5}Fe_2O_4$ nanofibers. Although as-spun nanofibers were shrunk by calcination, the morphology of isotropic fibers was maintained without breaking the microstructure.

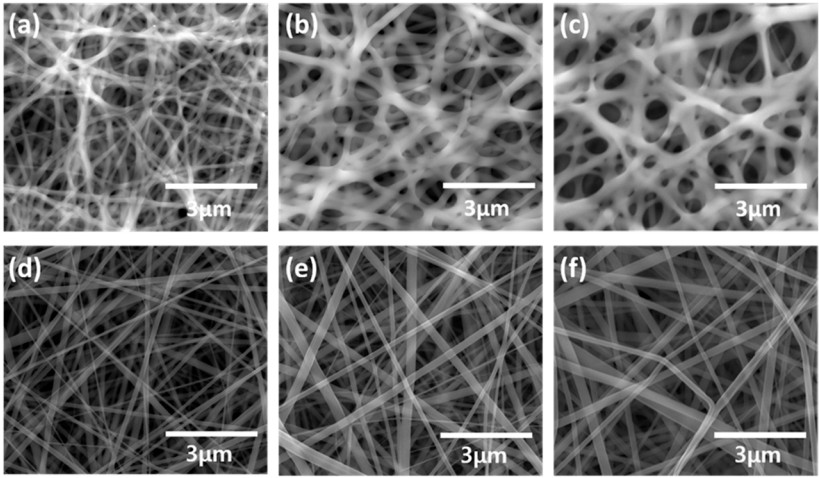

**Figure 3.** Field emission-scanning electron microscopes (FE-SEM) images of as-spun and calcined $Ni_{0.5}Zn_{0.5}Fe_2O_4$ nanofibers obtained from various samples: (**a**) As-spun nanofibers of sample A, (**b**) As-spun nanofibers of sample B, (**c**) As-spun nanofibers of sample C, (**d**) Calcined $Ni_{0.5}Zn_{0.5}Fe_2O_4$ nanofibers of sample A, (**e**) Calcined $Ni_{0.5}Zn_{0.5}Fe_2O_4$ nanofibers of sample B, and (**d**) Calcined $Ni_{0.5}Zn_{0.5}Fe_2O_4$ nanofibers of sample C.

Figure 4 shows the EDX mapping results of the nanofiber samples. Figure 3a, Figure 3b, and Figure 3c show the emission distribution of the characteristic X-ray of Fe, Ni, and Zn, respectively. A comparison of the distribution of each element with the SEM image shown in Figure 4d, confirmed that each component was evenly distributed at the position where the nanofiber existed. The density of each pixel was only related to the degree of the nanofiber overlap. The uniform distribution of each element can support XRD analysis results showing a single crystal structure. The EDX spectrum analysis of the mapping results is shown in Table 3. Due to the characteristics of the electron beam source, high analytical reliability for the magnetic powders cannot be expected, but the chemical composition ratio was very similar to the $Ni_{0.5}Zn_{0.5}Fe_2O_4$ phase.

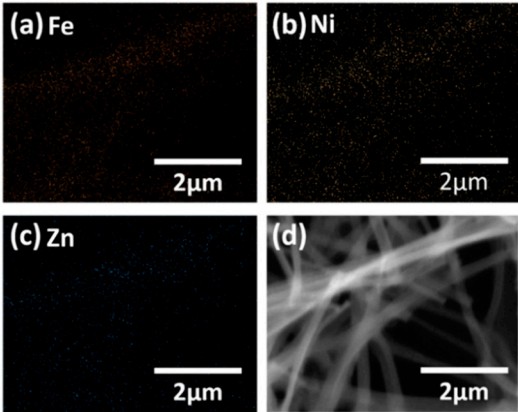

**Figure 4.** Energy dispersive X-ray spectroscopy (EDX) elemental mapping of $Ni_{0.5}Zn_{0.5}Fe_2O_4$ nanofibers: (**a**) Fe, (**b**) Ni, (**c**) Zn, and (**d**) FE-SEM image.

**Table 3.** Energy dispersive X-ray spectroscopy (EDX) spectrum analysis results of $Ni_{0.5}Zn_{0.5}Fe_2O_4$ nanofibers.

| Element | Weight Percent (%) | Atomic Percent (%) | Net Intensity | Error (%) |
|---------|---------|---------|---------|---------|
| Fe | 45.86 | 28.07 | 270.80 | 1.93 |
| Ni | 12.59 | 7.33 | 55.70 | 3.08 |
| Zn | 14.98 | 7.83 | 49.50 | 3.08 |
| O | 26.57 | 56.77 | 0 | 0 |

The diameters of $Ni_{0.5}Zn_{0.5}Fe_2O_4$ nanofibers were measured with FE-SEM images. To obtain quantification uniformity, 10 specimens for each sample A, B, and C were prepared and FE-SEM analysis was conducted. Four FE-SEM images were taken from each specimen and 5 points in the image with up, down, left, right, and center parts, were selected to measure the diameter. Figure 5 shows a histogram on the distribution of nanofiber diameters obtained by making 200 measurements of samples A, B, and C. The histogram shows that the average diameter of the samples was formed with a constant difference of about 20 nm within the range of 224 to 265 nm. This indicates that the diameter of the final oxide nanofibers could be controlled by adjusting the polymer content. Table 4 shows the average, maximum, minimum, and standard deviation of the diameters of each sample.

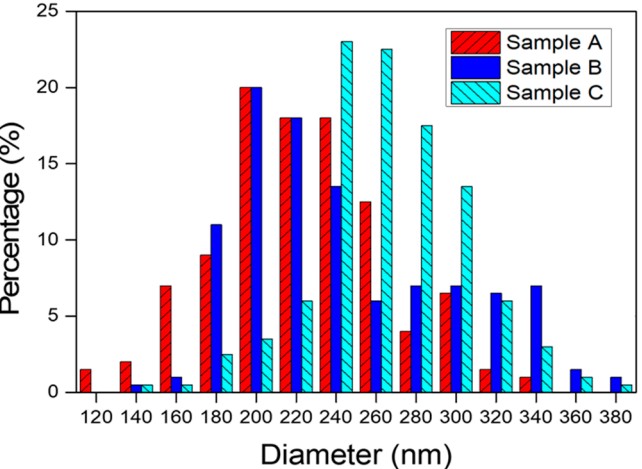

**Figure 5.** Histogram of the size distribution of $Ni_{0.5}Zn_{0.5}Fe_2O_4$ nanofibers.

**Table 4.** Maximum, minimum, and average diameters, and standard deviations of $Ni_{0.5}Zn_{0.5}Fe_2O_4$ nanofibers measured from samples A, B, and C.

| | $Ni_{0.5}Zn_{0.5}Fe_2O_4$ Nanofibers | | | |
|---|---|---|---|---|
| **Samples** | **Maximum Diameter (nm)** | **Minimum Diameter (nm)** | **Average Diameter (nm)** | **Standard Deviation (nm)** |
| Sample A | 355.02 | 112.9 | 224.26 | 45.42 |
| Sample B | 391.68 | 139.66 | 245.48 | 55.78 |
| Sample C | 402.52 | 157.54 | 266.42 | 39.87 |

Figure 6 shows the magnetization behavior of nanofiber samples with different average diameters measured by VSM. In general, the crystalized structure, chemical composition, oxygen vacancies, particle shape, and the interaction of the domain wall with various defects are considered to be the parameters of saturation magnetization [40–42]. Among them, the presence of non-aligned surface spin derived to thermal fluctuation is regarded as the main factor to reduce saturation magnetization, and it is related to the specific surface area [43]. In general, specific surface area is considered the main factor of the magnetization of nanoparticles due to the non-alignment of thermally fluctuating spin. Since one-dimensional nanofibers are a continuum of the smaller crystals, the influence of crystallite size and diameter must be considered. It is known that the saturation magnetization ($M_s$) increases with the crystallite size of a crystal [44]. The crystallite sizes of our samples were estimated by the Scherrer equation. The crystallite sizes of samples A, B, and C were 20 nm, 16 nm, and 13 nm, respectively. According to this, the $M_s$ of sample A is the largest and should be gradually decreased in the order of samples B and C, but samples B and C were almost at the same levels. The saturation magnetization can be affected by several parameters, such as calcination temperature, crystal size, and crystal structure. Herranz et al. [45] reported that Co-doped LaSrTi oxide films obtained at low oxygen pressures had a higher $M_s$ value than those obtained at high oxygen pressure. Sample C, which had the highest PVP content, must burn more PVP during the calcination process. PVP combustion decreases the oxygen pressure, which causes more oxygen vacancies in the crystalized structure in the formation process. In sample C, the decrease in $M_s$ resulting from the small crystallite size can be compensated by the oxygen vacancies effect. The concentration of the oxygen vacancy in the samples will be measured and discussed in a future study.

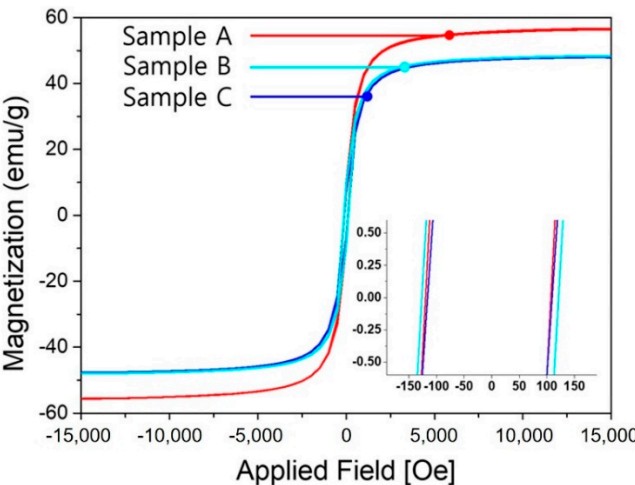

**Figure 6.** Magnetization versus magnetic field curves (M-H hysteresis loops) of $Ni_{0.5}Zn_{0.5}Fe_2O_4$ nanofibers.

To verify the values of the magnetic property of samples A, B, and C, the saturation magnetization, coercivity, and remnant magnetization of each sample were summarized and compared with previous reports on NiZn ferrite nanofibers and nanoparticles. The results are shown in Table 5 [8,46–49].

Although the range of saturation magnetization and coercivity change varies depending on the chemical composition, such as the content of Ni and Zn ions and the heat treatment temperature, the values of samples A, B, and C were in the general range, which means that samples A, B, and C were properly fabricated by electrospinning and the values of the magnetic property could be controlled. In the previous reports [8,46–49], the saturation magnetization and coercivity of NiZn ferrite nanofibers tended to be larger than in nanoparticles, which is thought to be due to the shape magnetic anisotropy and the small influence of thermal fluctuations.

**Table 5.** Saturation magnetization, coercivity, and remnant magnetization of $Ni_{0.5}Zn_{0.5}Fe_2O_4$ nanofibers measured from samples A, B, and C.

| Samples | Saturation Magnetization (emu/g) | Coercivity (Oe) | Remnant Magnetization (emu/g) |
|---|---|---|---|
| Sample A | 56.65 | 153.12 | 9.82 |
| Sample B | 48.09 | 150.41 | 7.37 |
| Sample C | 48.29 | 169.45 | 9.32 |
| NiZn ferrite Nanofibers [8,46,47] | 37–90.2 | 43–175.3 | - |
| NiZn ferrite Nanoparticles [48,49] | 23.95–56 | 16–200 | - |

## 4. Conclusions

$Ni_{0.5}Zn_{0.5}Fe_2O_4$ ferrite nanofibers were fabricated by electrospinning with a precursor solution of PVP, DMF, metal nitrate, and heat treatment. Since the nanofibers have a smaller specific surface area than the 0-dimensional nanoparticles, we confirmed that the diameter could not be a major factor of $M_s$ value when the difference of the samples was not large. Rather, a large amount of polymer added for the diameter control decreased the crystallite size during the calcination process and caused an unexpected change in the $M_s$ value. On the other hand, we confirmed that, when the diameter difference is not significant, the crystallite size can be a factor leading to the $M_s$ value tendency. The ability to tune the constants related to magnetic forces when making a film for electromagnetic interference shielding with a magnetic ceramic filler means that additional channels are available for optimizing the shielding efficiency and the frequency band. Moreover, the change of the crystallite size, according to the polymer content, provides a clue to controlling the magnetization behavior more economically than the conventional method with the heat treatment temperature control. It is possible to obtain the maximum shielding effect at the same cost by applying a combination of research themes of the size-dependent magnetic characteristics, development of a ferrite composition with excellent magnetic characteristics, and study of thickness to maximize impedance matching.

**Author Contributions:** Conceptualization, K.-H.N. and W.-Y.C. Methodology, K.-H.N. Validation, K.-H.N. and W.-T.K. Formal analysis, K.-H.N. Investigation, K.-H.N. and W.-T.K. Resources, T.-H.S. and W.-Y.C. Data curation, K.-H.N. Writing—original draft preparation, K.-H.N. Writing—review and editing, W.-Y.C. Visualization, K.-H.N. and W.-T.K. Supervision, W.-Y.C. Project administration, T.-H.S. and W.-Y.C. Funding acquisition, T.-H.S. and W.-Y.C.

**Funding:** The Korea Agency for Infrastructure Technology Advancement under Construction Technology R&D project (Grant No. 18SCIP-B146646-01) and National Research Foundation of Korea (Grant No. 2019R1I1A3A01057765) supported this work.

**Acknowledgments:** The authors thank the researchers (Sung-Wook Kim) of the Korea Institute of Civil Engineering and Building Technology, for their time and contributions to the study.

**Conflicts of Interest:** The authors declare no conflict of interest.

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
