# Peer review of "Magnetic Properties of NiZn Ferrite Nanofibers Prepared by Electrospinning"

_applsci, doi:10.3390/app9204297_

Round 1

Reviewer 1 Report

Title: Magnetic properties of NiZn ferrite nanofibers prepared by electrospinning

The authors describe a way to obtain nanofibers composed of NiZn ferrite through the electrospinning of a polymeric solution containing oxide precursors.

The polymer exploited to produce the nanofibers was PVP, while the NiZn ferrite fibers were obtained after treating the spun mats at 750°C for 4 hours.

Electrospinning represent a processing technique which allows to produce fibers from a polymer solution. In a proper set-up the jet should reach the collector dried from the solvent, resulting in a well-defined fiber deposition. The authors report the need of drying the fibers in oven after the deposition to let the DMF to completely evaporate from the spun mat. The presence of solvent in the spun sample is evident from the SEM, in which bad quality fibers are reported. The authors should improve the optimization of the processing conditions in order to get better quality fibrous morphologies.

Since both the polymer and the oxide precursors are soluble in water why don’t choose water as solvent which has lower boiling point and it’s green?

The weigh loss step observed below 100°C is likely related to the loss of the only moisture, since the boiling point of DMF is 153°C.

After the thermal treatment the fibers present a completely different morphology if compared to the spun mat. They resemble more to polymer fibers rather than oxide fibers, especially if compared to a recent work of the same group (Thin Solid Films 660 (2018) 358-364). Is there any possibility that the uploaded image reports mistakenly SEM belonging to different samples?

Besides, the decrease of the mean diameter related to the polymer degradation and volatilization is not visible in the calcined samples.

I would suggest also TEM characterization in order to confirm the results obtained from Scherrer equation related to grains size.

Author Response

An answer file for the comments was attached.

Reviewer 2 Report

Please see the attached files. Thanks! 

Author Response

(The authors gave the same response as above.)

Reviewer 3 Report

The paper by Na et al. described the Magnetic properties of NiZn ferrite nanofibers prepared by electrospinning; this work is a good contribution to the field and could be published in Applied Sciences after major revision as mentioned below:

The most recent review in this field should be cited in the introduction and discussed. See for instance Applied Materials Today, 2019, 17, 1-35 English is very poor. It should be checked by a native speaker; In addition, the paper contains a lot of typographical errors. Please read carefully and correct XRD could determine crystallite size and not grain size. Please correct CAs number, purities and provider of all chemicals should be given in the experimental sections Error bar should be added to all values, tables and figures given in the paper; is it relevant for instance to add 2 number after the decimal points for crystallite size, EDx measurement of diameter determined by SEM? Why the total of atomic percent or Weight percent is not equal to 100 in EDX? Conclusion should be rewritten and focused on the content of the manuscript; lattice strain was mentioned for instance just in the conclusion and not discussed before in the paper

Author Response

(The authors gave the same response as above.)

Round 2

Reviewer 1 Report

The main topic of the work is the production of oxide nanofibers obtained via thermal treatment of an oxide precursor-containing polymer spun mat.

Regarding the production of nanofibers via electrospinning and subsequent thermal treatment, the description of the results still lack clarity.

In the first round the authors have been asked to improve the fibrous morphology of the spun mat. In the present form the results reported are far from being ascribed as oxide precursor-containing polymer nano-fibers, since it is hardly recognizable a fibrous morphology. The spun mat does not present well defined fibers, but layers of spun product melted together instead, probably due to either a not complete evaporation of the solvent during the time of flight or a low polymer concentration. Furthermore, for this reason is not possible to calculate a good diameter distribution to compare with that of calcined fibers.

In the reported TGA the authors state the presence of a residue corresponding to the 12% of the initial weight. This residue has been related to NiZn ferrite spinel. If so, the calcined fibers should be characterized by a high shrinking in the mean diameter due to the loss of the 88% of the initial weight. This feature is not visible in the reported SEM. Besides, the calcined SEM morphologies do not fit with that of oxide fibers obtained via pyrolysis of oxide precursor-containing polymer matrix.

Based on the completely different morphology from the spun to the calcined fibers it is hard to believe that one is the product of the other. The authors state that the oxide precursors were completely dissolved together with PVP in DMF, meaning a homogeneous distribution of the precursor within the polymer solution. If so, after the thermal treatment, the morphology should resemble to that of the spun mat. This feature is missing.

Author Response

(The authors gave the same response as above.)

Reviewer 3 Report

Authors addressed all my comments. Paper could be published now in Applied Sciences

Author Response

(The authors gave the same response as above.)
